# European Prevalence of Polypoidal Choroidal Vasculopathy: A Systematic Review, Meta-Analysis, and Forecasting Study

**DOI:** 10.3390/jcm11164766

**Published:** 2022-08-16

**Authors:** Elon H. C. van Dijk, Jeppe K. Holtz, Marc J. Sirks, Janni M. E. Larsson, Roselie M. H. Diederen, Reinier O. Schlingemann, Camiel J. F. Boon, Yousif Subhi

**Affiliations:** 1Department of Ophthalmology, Leiden University Medical Center, 2333 ZA Leiden, The Netherlands; 2Department of Ophthalmology, Odense University Hospital, 5000 Odense, Denmark; 3Department of Otolaryngology, Odense University Hospital, 5000 Odense, Denmark; 4Department of Ophthalmology, Amsterdam University Medical Centers, University of Amsterdam, 1012 WX Amsterdam, The Netherlands; 5Department of Ophthalmology, Rigshospitalet, 2600 Glostrup, Denmark; 6Department of Ophthalmology, Jules Gonin Eye Hospital, Fondation Asile Des Aveugles, University of Lausanne, 1015 Lausanne, Switzerland; 7Department of Clinical Research, University of Southern Denmark, 5230 Odense, Denmark

**Keywords:** polypoidal choroidal vasculopathy, prevalence, Europe, systematic review, meta-analysis, forecasting

## Abstract

The purpose of this study was to provide an estimate of the number of current and future patients with polypoidal choroidal vasculopathy (PCV) in Europe. We systematically searched 11 literature databases on 18 May 2022 for studies on the prevalence of PCV among a consecutive and representative group of patients with suspected neovascular age-related macular degeneration (AMD). Prevalence of PCV in patients with suspected neovascular AMD was summarized and included in a prevalence meta-analysis. We then used current population data and population forecasts by Eurostat and the Office for National Statistics to determine current and future number of patients with neovascular AMD in Europe. Then, we calculated the number of patients with PCV with our calculated estimate of the prevalence of PCV among Europeans suspected with neovascular AMD. A total of five eligible studies were identified which included a total of 1359 patients. All these studies used the gold standard of indocyanine green angiography as a routine part of their diagnostic approach. Among patients undergoing detailed retinal examination for suspected neovascular AMD, our meta-analysis calculated the prevalence of PCV to be 8.3% (95% confidence interval: 6.8–9.8%). Our population estimates find that a total of 217,404 patients with PCV exist in Europe in the year 2022, which constitutes 0.04% of the entire population of Europe. This number is estimated to increase to 287,517 patients in the year 2040. Our estimates are important for different healthcare stakeholders, especially when planning and allocating expensive resources.

## 1. Introduction

Polypoidal choroidal vasculopathy (PCV) was originally described by Yannuzzi in 1982 as a variant of exudative age-related macular degeneration (AMD) [1]. Since then, numerous clinical reports have described this condition in more detail [2,3,4,5]. PCV is a chorioretinal disease with vascular aneurysmal polyp-like lesions with or without an associated branching vascular network, most probably originating from the inner choroid [6]. This vasculopathy leads to protrusion through Bruch’s membrane to the sub-retinal pigment epithelium (RPE) space with exudation into the subretinal space and the neuroretina. The term ‘polyp’ is actually a misnomer, as the polypoidal lesions may be an aneurysmal dilation of the neovascular network [6]. Hence, some authors advocate the use of the term ‘aneurysmal type 1 neovascularization’ [6]. PCV may encompass a spectrum of clinical and pathophysiological subtypes [7,8]. Van Dijk et al. have described three clinical subtypes of PCV: type A PCV (PCV-AMD), which is phenotypically and presumably pathophysiologically more associated with neovascular AMD and drusen; type B PCV, in which PCV is associated with a branching vascular network of non-PCV neovascularization but without drusen (PCV-BVN); and type C PCV, in which patients have a polyp-like lesion without a branching vascular network and without associated signs of AMD such as drusen [7]. Whereas type A PCV, similar to AMD, appears to be associated with a normal to thin choroid, a sizeable subgroup of types B and C may be associated with a normal to thick choroid (pachychoroid) [7]. Many cases also present with subretinal hemorrhage [6]. The gold standard of PCV diagnosis includes indocyanine green angiography (ICGA), which can reveal sub-RPE structures in detail, reveal polyp-like choroidal vascular lesions, and in many cases, an associated branching vascular network [9]. Patients with PCV who do not receive treatment are at risk of fibrovascular scarring and damage to the neuroretina with potentially severe visual impairment [10,11]. Treatment with intravitreal injections of anti-vascular endothelium growth factor (anti-VEGF) medication and photodynamic therapy (PDT) have been found to lead to reduced exudation and closure of polyp-like vascular lesions, which often preserve relatively good vision [12,13,14,15,16].

The pathophysiology of PCV remains incompletely understood. Some forms of PCV have been linked to venous stasis in the choroid with secondary choroidal anastomoses and choroidal vascular remodeling [17]. In line with this hypothesis, some patients with PCV have pachychoroid features upon choroidal imaging and often present without drusen maculopathy [18,19]. This is also supported by studies in which patients with PCV were shown not to have age-related immunological changes that are otherwise associated with AMD [20,21]. On the other hand, another subgroup of PCV patients (e.g., type A PCV according to Van Dijk et al.) may have a pathophysiology that is more similar to that of AMD [22].

Studies on PCV, especially epidemiological population-based and registry-based studies, have been challenged by the lack of a separate International Classification of Diseases (ICD) diagnosis code for the disease. There are currently no good estimates of the prevalence of PCV, which is further challenged by the fact that the disease in Europe is often not recognized. Approximately half of patients with suspected neovascular AMD are diagnosed with PCV in Asians, whereas this is only 8–9% among whites [19,23]. The unknown estimate of the disease burden is a challenge for various reasons, e.g., when planning national health service or when applying for funding for PCV research, as it is not clear how many patients suffer from this disease. Further, the lack of an estimation of the disease burden can also hinder further political attention to the worldwide shortage of verteporfin (Visudyne^®^, Cheplapharm Arzneimittel GmbH, Greifswald, Germany) which is used for PDT [24], and therefore, to a large extent, currently unavailable for patients with PCV [25].

In this study, we address this issue by providing the first European prevalence estimate of PCV. Since no population-based prevalence estimate of PCV exists, our approach is to systematically review the literature on the prevalence of PCV among European patients suspected of neovascular AMD and calculate a summary estimate using a prevalence meta-analysis. We then apply this estimate to the best current estimate of neovascular AMD in Europe, and to the most likely scenario of population forecast of European countries as provided by the Eurostat and the Office for National Statistics.

## 2. Materials and Methods

### 2.1. Study Design

This study consisted of three stages. The first stage was a systematic review and meta-analysis of the prevalence of PCV in patients suspected with neovascular AMD. The second stage was to use age-stratified prevalence estimates of neovascular AMD and apply them to current age-stratified population statistics of countries in Europe and to similar age-stratified estimated future population statistics of countries in Europe. In the third and final stage of this study, we used our calculated prevalence estimate of PCV in patients suspected with neovascular AMD in a European population and applied this estimate on the current and future prevalence estimates of neovascular AMD in Europe. The systematic review was reported according to the recommendations of the Preferred Reporting Items for Systematic Reviews and Meta-Analyses (PRISMA) and the protocol was registered a priori in the PROSPERO database (no. CRD42022334049). According to Danish and Dutch law, institutional review board approval is not relevant for systematic reviews nor for forecasting studies of publicly available population data nor future estimates.

### 2.2. Eligibility Criteria

Eligible studies were defined as those which evaluated the prevalence of PCV in patients suspected with neovascular AMD. The study had to be performed in an European population. We did not restrict diagnostic modalities employed for diagnosing PCV or neovascular AMD, nor restricted diagnostic criteria for PCV or neovascular AMD, but noted the author’s definitions of these aspects. For a study to be eligible, we required that data on PCV should be reported on the patient level. The population had to be representative of the broad population of patients with neovascular AMD and not pre-selected for a certain reason, e.g., poor-responders on anti-VEGF therapy or only type 1 macular neovascularization. Studies were expected to be observational in nature, but we did not restrict on this definition and allowed relevant data from any study design. Single case studies, publications without original data, conference abstracts, or animal studies were not considered eligible. For practical purposes, we only considered studies disseminated in the English language.

### 2.3. Information Sources, Literature Search, and Study Selection

We searched the literature databases PubMed, EMBASE, Web of Science Core Collection, BIOSIS Previews, Current Contents Connect, Data Citation Index, Derwent Innovations Index, KCI-Korean Journal Database, SciELO Citation Index, and the Cochrane Central. One trained author (Y.S.) conducted the search on 18 May 2022. Details of the search phrases tailored to the individual literature databases are available in Appendix A. One author (Y.S.) examined the title and abstract of all identified records and removed duplicates and those deemed obviously irrelevant. Remaining references were retrieved in full text for evaluation of eligibility. Two authors (J.H. and J.M.E.L.) independently examined these full text studies as well as references from these studies for any additional relevant studies. Disagreements between the authors were discussed and in the lack of consensus, a third author (Y.S.) made the final decision.

### 2.4. Data Collection Process and Risk of Bias of Individual Studies

Data on study characteristics, population characteristics, methods for diagnosis, and results were extracted from each study using pre-designed data extraction forms. We anticipated that most studies would be cross-sectionally designed and therefore evaluated risk of bias of individual studies using the relevant items from the Agency for Healthcare Research and Quality (AHRQ) checklist for Cross-Sectional Studies (Questions 1–4 and 6), which is the recommended tool for evaluating cross-sectional studies [26]. Two authors (J.H. and J.M.E.L.) independently extracted data and evaluated risk of bias of individual studies. Disagreements between the authors were discussed and if consensus could not be reached, a third author (Y.S.) made the final decision.

### 2.5. Outcomes and Summary Measures, Synthesis of Results, and Risk of Bias across Studies

The primary outcome measure was the prevalence of PCV in eyes suspected with neovascular AMD. Our unit of analysis was per patient since this is also the unit of analysis for patient prevalence in the following steps of our study. Meta-analysis was performed using MetaXL 5.3 (EpiGear International, Sunrise Beach, QLD, Australia) for Microsoft Excel 2013 (Microsoft, Redmont, WA, USA). The random-effects model was employed to account for potential heterogeneity across studies. Caution must be exercised in prevalence meta-analyses when a number reaches the extremes (i.e., 0% or 100%) since this can result in variance instability and erroneous weighting of studies [27]. To accommodate to this potential issue, all prevalence numbers were transformed for analysis using the double arcsine method and were then back transformed for interpretation [27]. Heterogeneity was evaluated using the Cochran’s Q and I^2^ [28]. A Funnel plot was used to evaluate any skewed results and publication bias [29]. The final summary measure was the prevalence estimate of PCV. Sensitivity analysis was conducted by removing each study in turn and re-calculating the summary measure to evaluate the magnitude of the change in the results.

### 2.6. Prevalence Estimation and Forecasting Analysis

Li et al. estimated the prevalence of AMD in Europe in a systematic review and meta-analysis [30]. Based on their meta-analysis on 55,323 European individuals, the authors calculated the prevalence of neovascular AMD to 0.1% (95% confidence interval (CI): 0.1 to 0.3%) for individuals aged ≤64 years, 0.8% (95% CI: 0.6 to 1.0%) for individuals aged 65–74 years, and 3.3% (95% CI: 2.5 to 4.2%) for individuals aged ≥75 years. These prevalence estimates represent, to our knowledge, the best current and highest level of evidence on the prevalence of neovascular AMD in Europe. These estimates were used for the following steps in our study.

We extracted publicly available data on country population statistics and the most likely population project scenario from Eurostat (Austria, Belgium, Bulgaria, Croatia, Cyprus, Czech Republic, Denmark, Estonia, Finland, France, Germany, Greece, Hungary, Iceland, Ireland, Italy, Latvia, Liechtenstein, Lithuania, Luxembourg, Malta, the Netherlands, Norway, Poland, Portugal, Romania, Slovakia, Slovenia, Spain, Sweden, Switzerland) and from the Office for National Statistics (United Kingdom defined as the combination of England, Northern Ireland, Wales, and Scotland). Countries included from this approach include almost the entirety of Europe by various geographical and political definitions.

Population data was stratified according to individuals aged ≤64 years, 65–74 years, and ≥75 years and summarized in Appendix A. We used these age stratified data to calculate the current and estimated future number of patients with neovascular AMD in Europe. Afterwards, we used our calculated prevalence summary estimate to calculate the proportion of these patients with neovascular AMD, which can be assumed to have PCV upon further examination. Then, we could estimate the prevalence of PCV.

## 3. Results

### 3.1. Literature Search and Study Selection

Our literature search identified a total of 744 records. We then discarded duplicates (*n* = 280) and records obviously irrelevant (*n* = 453). The remaining 11 records were evaluated in full text. Of these, six records were excluded as they did not fulfill our eligibility criteria (Figure 1), and we included five studies for qualitative and quantitative review.

### 3.2. Study and Population Characteristics

The five eligible studies for review included a total of 1359 patients [10,31,32,33,34]. All studies were retrospective in nature, cross-sectionally designed, and performed in a single center. Studies originated from Denmark (*n* = 2), Greece (*n* = 1), Italy (*n* = 1) and the United Kingdom (*n* = 1). All studies described that the patients underwent fundus examination, fluorescein angiography, and ICGA. Two studies also described optical coherence tomography as part of their examination [10,30]. Details regarding the study characteristics and eligibility criteria are outlined in Table 1.

Population demographics were similar across groups as well as across studies (Table 2). The mean age ranged between 70–77 years in patients with PCV and between 73–79 years in patients with neovascular AMD. Females constituted between 41–65% of patients with PCV and 49–68% of patients with neovascular AMD. Diagnostic definitions of PCV were described in detail in four studies [10,32,33,34] and one study simply described classical findings on ICGA and referred to the early studies of Yannuzzi for details [31]. Diagnostic definition of neovascular AMD was only described in one study [10].

### 3.3. Results and Risk of Bias of Individual Studies

Ilginis et al. found that 8% of patients with suspected neovascular AMD were found to actually have PCV upon further examination in a Scandinavian population [31]. Another Scandinavian population was described by Lorentzen et al., who reported that 6% of their patients with suspected neovascular AMD had PCV [10]. This study also reported that the majority of cases were hemorrhagic at presentation [10]. Yadav et al. reported on their findings from United Kingdom, where PCV was identified in 9% of the patients [34]. The authors also reported that PCV was only found in eyes with type 1 macular neovascularization and that PCV constituted 22% of such type 1 macular neovascularization [34]. Ladas et al. and Scassellati-Sforzolini et al. reported the prevalence in Southern European patient populations [32,33]. In Greece, the prevalence of PCV was 8%, was located mostly in the peripapillary area, and fellow eyes were less likely to have large drusen (20%) when compared to eyes diagnosed with neovascular AMD (81%) [32]. In Italy, similar patterns were reported of PCV having a higher incidence of extrafoveal presentation and fewer drusen in the fellow eye when compared to patients with neovascular AMD [33].

Risk of bias evaluation of individual studies showed that all studies clearly defined the source of data and performed consecutive recruitment, and that most studies clearly defined eligibility criteria, time period of participant recruitment/eligibility, and quality assurance protocol. Exclusions were explained clearly in one study, and not clearly in two studies, and not explained in two studies. Details of the risk of bias of individual studies are listed in Table 3.

### 3.4. Synthesis of Results and Risk of Bias across Studies

Synthesis of results using the random-effects model showed a pooled prevalence of PCV in patients suspected with neovascular AMD of 8.3% (95% confidence interval: 6.8–9.8%) (Table 4). Heterogeneity across studies was insignificant and quantified as I2 = 0.2 and Cochran’s Q = 4.0. The Funnel plot did not suggest a significant presence of risk of bias across studies (Appendix A). The sensitivity analysis showed robustness of the summary estimate as excluding studies in turn only led to minor changes; the summary estimate only varied between 7.7 and 9.0% (Appendix A).

### 3.5. Estimated Current and Future Number of Patients with PCV in Europe

We extracted statistics on country-specific population and forecast of population development for years 2022, 2025, 2030, 2035, and 2040 within an age strata of ≤ 64 years, 65–74 years, and ≥75 years (Appendix A). These numbers were multiplied with the prevalence of neovascular AMD within each age stratum as estimated by Li et al. [24]. The resulting estimate of current and future numbers of patients with neovascular AMD are summarized in Appendix A. Briefly, we estimate that 2.6 million individuals in Europe have neovascular AMD in 2022 and that this number is expected to increase gradually to 2.8 million, 3.0 million, 3.2 million, and 3.5 million, respectively, in the years 2025, 2030, 2035, and 2040. Based on these numbers, and our calculated estimate that 8.3% (95% confidence interval: 6.8–9.8%) of patients with neovascular AMD upon further examination will have PCV, we were able to calculate that 217,404 (95% confidence interval: 178,114–256,694) individuals have PCV in Europe in 2022. This number is expected to gradually increase to 287,517 (95% confidence interval 235,556–339,478) in the year 2040. Country-specific and total prevalence estimates of PCV as well as estimated change rates over time are all summarized in Table 5.

## 4. Discussion

In this systematic review and meta-analysis of a total of 1359 European patients with suspected neovascular AMD, the prevalence of PCV is estimated to be 8.3% (95% confidence interval: 6.8–9.8%). Based on this estimate, and our estimate of neovascular AMD in Europe, we calculate that there are approximately 220,000 patients with PCV in Europe today, a number expected to grow to approximately 300,000 in 2040. Although this number is much smaller than that of neovascular AMD [30], or other highly prevalent ophthalmic diseases in Europe [35], this is still a significant number of patients also in terms of disease and treatment burden, constituting 0.04% of the entire population of 531 million Europeans. These estimates are important for different healthcare stakeholders, especially when planning and allocating resources.

Moreover, the current study addresses the importance on the early diagnosis of PCV, as misdiagnosis can negatively affect the prognosis, and tailored treatment—which usually consists of a combination of PDT and anti-VEGF injections—is therefore required in PCV. Ophthalmologists should have a high index of suspicion and low threshold to perform additional imaging such as combined fluorescein angiography/ICGA in the case of signs suggestive of PCV. Such signs include a pink-orange subretinal nodular lesion on fundoscopy; a peaked RPE elevation on optical coherence tomography scanning, often with a hyperreflective subretinal accumulation beneath it; and either non-response or partial response to anti-VEGF treatment in patients with a neovascularization without evidence of another diagnosis other than AMD [8,36].

Previous studies suggest that white patients with PCV are on average 3.7 years younger than patients with neovascular AMD and that no significant gender difference exists, which could have led to a source of bias in the current study [19]. In particular, the increase in the number of patients with neovascular AMD is largely attributed to the growth of number of elderly individuals in the population [37,38,39]. If PCV pathogenesis is different from neovascular AMD [19,20,21], one can question if the number of patients with PCV will increase in a similar fashion to that of neovascular AMD. This could affect the accuracy of our forecast, also taking the three PCV subtypes into account that have been recently described. The number of type A PCV patients, which present with clinical features that are similar to neovascular AMD, may increase more than numbers of type B PCV and type C PCV patients [7].

Our study has several limitations. First, we rely our estimate on studies of entirely or almost entirely white populations, whereas immigration to Europe leads to an increase in the number of individuals with different ethnicities. This means that as the immigrants grow older, our estimates become less accurate. Second, our estimates are based on the current best forecast from the Eurostat and the Office for National Statistics. Unforeseen developments in economy, politics, immigration, emigration, and death may change the population numbers towards 2040. Thus, the estimates of 2022 can be considered more reliable than those of 2040. Third, we assume that the prevalence of PCV in patients suspected of nAMD will remain constant in future. These are assumptions that may or may not hold true; however, without such assumptions our calculations would not be possible. Finally, our estimates are only as accurate as the studies that we could use for data analysis. Our risk of bias within individual studies did not identify important sources of bias in the study design. Our Funnel plot was not suggestive of a strong risk of bias across studies. However, variation in the definition of PCV, referral pattern differences, poor image quality, and media opacity may all influence the prevalence estimates of the individual studies. Indeed, most of the studies in the meta-analysis include patient cohorts from tertiary referral centers. One can argue that referral pattern differences may be prone to a different prevalence of PCV when examined in tertiary centers. For example, an OCT appearance of type 1 CNV may lead a referral to a tertiary center as the referring ophthalmologist may argue that an ICGA may be needed for further examination. Thus, the proportion of PCV patients may be higher in tertiary referral centers than in the regular population. Calculating a population-wide prevalence from data from tertiary referral center studies possesses the risk of an overestimation of the actual PCV prevalence.

Taken together, we conclude that this study presents the first European prevalence estimate of PCV. Based on our study, we estimate that there are currently approximately 220,000 patients with PCV in Europe, which constitute 0.04% of the entire population of Europe. The number of patients is expected to grow at a rate of 1.6% per year.

## Figures and Tables

**Figure 1 jcm-11-04766-f001:**
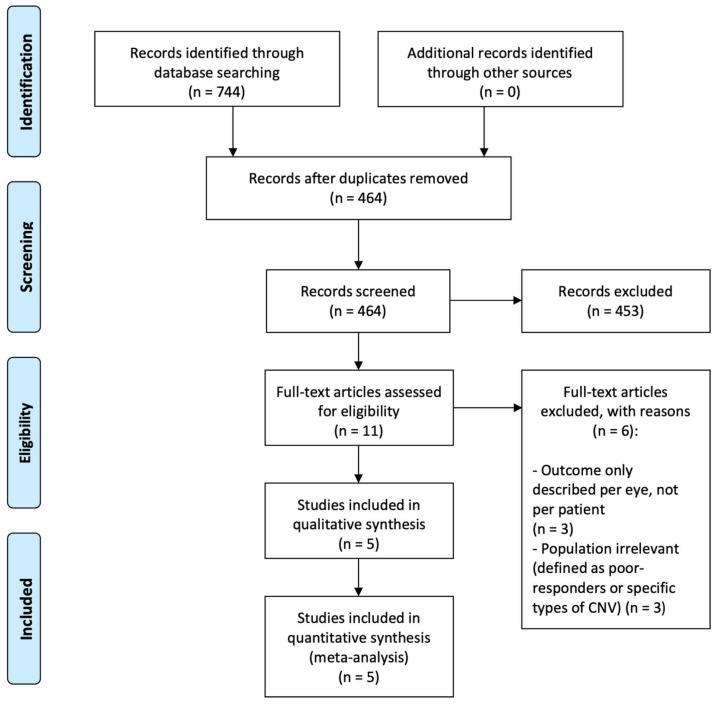
PRISMA flow diagram of study selection.

**Table 1 jcm-11-04766-t001:** Study characteristics of eligible studies.

Reference	Country	Study Design	Population Description	Clinical Examination Modalities
Ilginis et al., 2012 [31]	Denmark	Retrospective, cross-sectional, single-center	Consecutive patients with presumed classic CNV, occult CNV, or RAP referred to the clinic during a period of six months. Patients were excluded if aged <50 years, had inflammatory CNV, high myopia, or angioid streaks.	Fundus examination, FA, and ICGA
Ladas et al., 2004 [32]	Greece	Retrospective, cross-sectional, single-center	Consecutive patients with initial diagnosis of exudative AMD during a period of two years. Patients were aged >50 years and excluded if pathological myopia, presumed ocular histoplasmosis syndrome, or CSC.	Fundus examination, FA, and ICGA
Lorentzen et al., 2018 [10]	Denmark	Retrospective, cross-sectional, single-center	Consecutive patients referred for retinal diagnosis with presumed exudative AMD during a period of one year. Patients were excluded if not AMD or PCV, or if retinal angiography was not possible due to allergies.	Fundus examination, OCT, FA, and ICGA
Scassellati-Sforzolini et al., 2001 [33]	Italy	Retrospective, cross-sectional, single-center	Consecutive patients with presumed exudative AMD within 2 months from the onset of visual symptoms. Patients were excluded if aged <50 years, previous laser photocoagulation, any other retinal or choroidal disease apart from AMD/PCV including pathological myopia, angioid streaks, CSC, inflammation, presumed ocular histoplasmosis, and punctate inner choroidopathy.	Fundus examination, FA, and ICGA
Yadav et al., 2017 [34]	United Kingdom	Retrospective, cross-sectional, single-center	Consecutive patients with presumed neovascular AMD during a period of two years.	Fundus examination, OCT, FA, and ICGA

Abbreviations: AMD = age-related macular degeneration; CNV = choroidal neovascularization; CSC = central serous chorioretinopathy; FA = fluorescein angiography; ICGA = indocyanine green angiography; OCT = optical coherence tomography; PCV = polypoidal choroidal vasculopathy; RAP = retinal angiomatous proliferation.

**Table 2 jcm-11-04766-t002:** Population characteristics of eligible studies.

Reference	PCV			Neovascular AMD			Total
	*Clinical Definition*	*N*	*Age (Years) and* *Gender (Females)*	*Clinical Definition*	*N*	*Age (Years) and* *Gender (Females)*	*N*
Ilginis et al., 2012 [31]	Described as classical findings on ICGA, without any specific description of these findings.	7	77 ± 8 57%	N/A	82	78 ± 7 68%	89
Ladas et al., 2004 [32]	Presence of one or more polypoidal dilations in the inner choroid seen on ICGA as areas of early intense hyperfluorescence.	22	73 ± 8 41%	N/A	246	77 ± 7 49%	268
Lorentzen et al., 2018 [10]	Presence of one or more polyps seen in early-phase ICGA with a hypofluorescent halo with or without branching vascular network. Other characteristics were not mandatory but used to support the diagnosis: orange-red focal subretinal polyp-like structures in fundoscopy and retinal OCT with protrusion from the choroid that elevates the RPE from the Bruch’s membrane.	17	76 ± 8 65%	Presence of CNV membranes with occult CNV, classic CNV or RAP which present with serous detachment or intraretinal fluid.	282	79 ± 8 66%	299
Scassellati-Sforzolini et al., 2001 [33]	Presence of isolated or multiple polypoidal choroidal vasculopathy dilations with or without identifiable continuous branching choroidal vessels, all on ICGA seen as early intense hyperfluorescence.	19	70 ± 8 59%	N/A	175	73 ± 7 53%	194
Yadav et al., 2017 [34]	Presence of early subretinal hyperfluorescent lesions on ICGA and other features including nodular appearance of polyps on stereo images, hyperfluorescent halo around the nodule, pulsatile filling of polyps, branching vascular network, and orange appearance of nodules on color imaging corresponding to the ICGA.	45	75 ± 8 56%	N/A	447	79 ± 6 N/A	492

Age is summarized using mean ± standard deviation. Gender distribution is summarized using percentages of females. Abbreviations: AMD = age-related macular degeneration; CNV = choroidal neovascularization; ICGA = indocyanine green angiography; OCT = optical coherence tomography; PCV = polypoidal choroidal vasculopathy; RAP = retinal angiomatous proliferation; N = number.

**Table 3 jcm-11-04766-t003:** Risk of bias within individual studies included in the review.

Reference	Defines Source	Eligibility Criteria	Time Period	Consecutive Recruitment	Quality Assurance	Explains Exclusions
Ilginis et al., 2012 [31]	Yes	Yes	Yes	Yes	Yes	Yes
Ladas et al., 2004 [32]	Yes	Yes	Yes	Yes	Yes	No
Lorentzen et al., 2018 [10]	Yes	Yes	Yes	Yes	Yes	Unclear
Scassellati-Sforzolini et al., 2001 [33]	Yes	Yes	No	Yes	No	No
Yadav et al., 2017 [34]	Yes	No	Yes	Yes	Yes	Unclear

Studies are assessed on relevant items from the Agency for Healthcare Research and Quality checklist: Defines source: Defines the source of information. Eligibility criteria: Lists inclusion and exclusion criteria or refers to previous publications. Time period: Indicates time period used for identifying participants. Consecutive recruitment: Indicates whether or not subjects were consecutively recruited for the study. Quality assurance: Describes any assessments undertaken for quality assurance purposes. Explains exclusions: Explains any patient exclusions from analysis.

**Table 4 jcm-11-04766-t004:** Meta-analysis of the prevalence of polypoidal choroidal vasculopathy in patients suspected with neovascular age-related macular degeneration.

Reference	Prevalence	95%CI	Study Weight
Ilginis et al., 2012 [31]	7.9%	3.0–14.5%	6.7%
Ladas et al., 2004 [32]	8.2%	5.2–11.8%	20.0%
Lorentzen et al., 2018 [10]	5.7%	3.3–8.6%	22.3%
Scassellati-Sforzolini et al., 2001 [33]	9.8%	6.0–14.4%	14.5%
Yadav et al., 2017 [34]	9.1%	6.7–11.9%	36.6%
Pooled summary estimate	8.3%	6.8–9.8%	
*Heterogeneity statistics*	I^2^ = 0.2	Cochran’s Q = 4.0	

Abbreviations: 95% CI = 95% confidence interval.

**Table 5 jcm-11-04766-t005:** Estimated current and future prevalence of polypoidal choroidal vasculopathy in Europe.

Country			Year			Total Increase until 2040, %	Increase per Year until 2040, %
	2022	2025	2030	2035	2040		
Austria	3538	3728	4021	4417	4874	37.8%	1.8%
Belgium	4522	4773	5198	5667	6114	35.2%	1.7%
Bulgaria	2797	2894	3019	3097	3157	12.9%	0.7%
Croatia	1632	1688	1809	1930	2000	22.5%	1.1%
Cyprus	302	333	378	417	449	48.8%	2.2%
Czech Republic	4091	4399	4809	5005	5171	26.4%	1.3%
Denmark	2354	2516	2719	2887	3057	29.9%	1.5%
Estonia	535	559	596	633	666	24.5%	1.2%
Finland	2398	2605	2854	3005	3076	28.3%	1.4%
France	27,741	29,849	33,038	35,844	38,294	38.0%	1.8%
Germany	37,086	37,910	40,232	43,202	46,791	26.2%	1.3%
Greece	4874	5016	5251	5594	5915	21.4%	1.1%
Hungary	3706	3859	4166	4395	4472	20.7%	1.0%
Iceland	113	125	145	167	187	65.4%	2.8%
Ireland	1556	1709	1962	2220	2493	60.2%	2.7%
Italy	28,442	29,701	31,496	33,777	36,700	29.0%	1.4%
Latvia	777	787	804	837	870	12.0%	0.6%
Liechtenstein	15	16	19	21	24	63.2%	2.8%
Lithuania	1124	1140	1183	1266	1355	20.6%	1.0%
Luxembourg	199	216	246	283	324	62.4%	2.7%
Malta	195	220	250	276	296	52.2%	2.4%
Netherlands	6807	7405	8217	8970	9643	41.7%	2.0%
Norway	1956	2126	2379	2628	2847	45.5%	2.1%
Poland	13,229	14,338	16,231	17,867	18,534	40.1%	1.9%
Portugal	4596	4789	5141	5496	5884	28.0%	1.4%
Romania	7003	7227	7669	8244	8302	18.6%	1.0%
Slovakia	1760	1903	2183	2405	2568	45.9%	2.1%
Slovenia	839	888	992	1084	1157	37.9%	1.8%
Spain	19,577	20,702	22,556	25,019	27,830	42.2%	2.0%
Sweden	4280	4568	4902	5182	5456	27.5%	1.4%
Switzerland	3401	3622	3982	4383	4832	42.1%	2.0%
United Kingdom	25,960	27,555	29,623	31,778	34,180	31.7%	1.5%
Total	217,404	229,165	248,073	267,995	287,517	32.3%	1.6%
95% CI	178,114 to 256,694	187,750 to 270,581	203,240 to 292,905	219,562 to 316,428	235,556 to 339,478	8.3% to 56.2%	0.4% to 2.5%

Abbreviations: 95% CI = 95% confidence interval.

## Data Availability

Not applicable.

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
