# Peer review of "European Prevalence of Polypoidal Choroidal Vasculopathy: A Systematic Review, Meta-Analysis, and Forecasting Study"

_jcm, 2022, doi:10.3390/jcm11164766_

Round 1

Reviewer 1 Report

In this systematic review and meta-analysis, the authors summarized and estimated the prevalence of polypoidal choroidal vasculopathy (PCV) in subjects of neovascular age-related macular degeneration (nAMD) in the European populations. This study was conducted and reported following established guidelines. The results were meaningful for the field. The manuscript was well organized. I have some comments for the authors.

1. There were two assumptions when estimating the number of cases with PCV in European populations - (1) the occurrence of PCV in the nAMD is the same between age strata of ≤ 64 years, 65–74 years, and ≥ 75 years; and (2) the prevalence of PCV will stay unchanged in the future. These assumptions should be thoroughly discussed among other assumptions.

2. Figure 1 was not fully shown (missing connections between components).

3. It was unclear how the risk of bias assessments inform the data analysis and interpretation of the results.

4. All the lines in supplementary tables are missing.

5. Please add the total N of each study in Table 2. 

Author Response

Reviewer #1 general comments:

In this systematic review and meta-analysis, the authors summarized and estimated the prevalence of polypoidal choroidal vasculopathy (PCV) in subjects of neovascular age-related macular degeneration (nAMD) in the European populations. This study was conducted and reported following established guidelines. The results were meaningful for the field. The manuscript was well organized. I have some comments for the authors.

Authors’ response:

We thank the reviewer for the positive feedback and for the valuable comments.

Reviewer #1 comment #1:

  1. There were two assumptions when estimating the number of cases with PCV in European populations - (1) the occurrence of PCV in the nAMD is the same between age strata of ≤ 64 years, 65–74 years, and ≥ 75 years; and (2) the prevalence of PCV will stay unchanged in the future. These assumptions should be thoroughly discussed among other assumptions.

Authors’ response:

The reviewer is correct, these are indeed important assumptions to keep in mind when interpreting the results of this study. As proposed, we have now discussed these assumptions together with other assumptions and limitations of this study. 

Reviewer #1 comment #2:

  1. Figure 1 was not fully shown (missing connections between components).

Authors’ response:

Thank you for pointing this out. This has also been identified by Reviewer 2 and is likely to be a file transformation issue (transformation to PDF file). This has now been addressed.

Reviewer #1 comment #3:

  1. It was unclear how the risk of bias assessments inform the data analysis and interpretation of the results.

Authors’ response:

In this study, we have risk of bias evaluation within individual studies using the AHRQ checklist for cross-sectional studies, and a risk of bias evaluation across studies using the Funnel plot. Neither suggested a strong risk of bias, which is the reason that the discussion of risk of bias assessments is limited. Nevertheless, these issues are now discussed as suggested.

Reviewer #1 comment #4:

  1. All the lines in supplementary tables are missing.

Authors’ response:

We have now added lines to the supplementary tables as recommended.

Reviewer #1 comment #5:

  1. Please add the total N of each study in Table 2.

Authors’ response:

We have added the total N of each study in Table 2 as recommended.

Reviewer 2 Report

This is a well written manuscript contributing new thoughts about the epidemiology of PCV to the field. Congratulations to the authors. 

Some minor things need to be addressed:

Figure 1: The PRISMA flow diagram in its current form appears to be flawed, there is information missing that might have been lost in its transformation to PDF. Please revise figure 1.

The tables are a little crowded, more space between the lines would improve readability. This might a task for the editorial office as well.

Limitations: All studies included in the meta-analysis seem to be patient cohorts from tertiary referral centers. This could be a bias for epidemiological data because the more "severe" cases of AMD are referred while others might have their injections in lower order centers. As PCV is considered more aggressive and patients are younger, the proportion of PCV patients might be higher in tertiary referral centers than in the regular population. Calculating a population-wide prevalence from data from tertiary referral center studies might lead to an overestimation of the prevalence of PCV compared to population based epidemiological studies. I would kindly suggest the authors to clarify this is in the limitations section. 

Results l 258 and table 5:

The authors calculated confidence intervals for there prevalence estimate of PCV in the meta analysis. For some reason they refrained from calculating confidence intervals for their forecasting calculations. It makes sense not to overload table 5 but at least the total estimate should have confidence intervals for good scientific reporting. 

Author Response

Reviewer #2 general comments:

This is a well written manuscript contributing new thoughts about the epidemiology of PCV to the field. Congratulations to the authors. There are some minor things need to be addressed.

Authors’ response:

We thank the reviewer for the positive feedback and for the valuable comments.

Reviewer #2 comment #1:

Figure 1: The PRISMA flow diagram in its current form appears to be flawed, there is information missing that might have been lost in its transformation to PDF. Please revise figure 1.

Authors’ response:

Thank you for pointing this out. As the reviewer rightly suggests, this may be a file transformation issue. This has now been addressed.

Reviewer #2 comment #2:

The tables are a little crowded, more space between the lines would improve readability. This might a task for the editorial office as well.

Authors’ response:

We have now provided more space between the lines of the tables as recommended.

Reviewer #2 comment #3:

Limitations: All studies included in the meta-analysis seem to be patient cohorts from tertiary referral centers. This could be a bias for epidemiological data because the more "severe" cases of AMD are referred while others might have their injections in lower order centers. As PCV is considered more aggressive and patients are younger, the proportion of PCV patients might be higher in tertiary referral centers than in the regular population. Calculating a population-wide prevalence from data from tertiary referral center studies might lead to an overestimation of the prevalence of PCV compared to population based epidemiological studies. I would kindly suggest the authors to clarify this is in the limitations section.

Authors’ response:

Thank you for raising this issue. We have now raised this point as a potential source of bias and discussed this as a limitation as recommended by the reviewer.

Reviewer #2 comment #4:

Results l 258 and table 5:

The authors calculated confidence intervals for their prevalence estimate of PCV in the meta-analysis. For some reason they refrained from calculating confidence intervals for their forecasting calculations. It makes sense not to overload table 5 but at least the total estimate should have confidence intervals for good scientific reporting.

Authors’ response:

Thank you for this suggestion. As recommended, we have now provided confidence intervals of the total estimate in Table 5 as well as in the text.